# Associations between Mental Health, Lifestyle Factors and Worries about Climate Change in Norwegian Adolescents

**DOI:** 10.3390/ijerph191912826

**Published:** 2022-10-07

**Authors:** Marja Leonhardt, Marie Dahlen Granrud, Tore Bonsaksen, Lars Lien

**Affiliations:** 1Norwegian National Advisory Unit on Concurrent Substance Abuse and Mental Health Disorders, Innlandet Hospital Trust, 2381 Brumunddal, Norway; 2Faculty of Social and Health Sciences, Inland Norway University of Applied Sciences, 2418 Elverum, Norway; 3Department of Health, VID Specialized University, Campus Stavanger, 4024 Stavanger, Norway

**Keywords:** climate change, adolescents, worry, mental health, lifestyle factors, survey

## Abstract

Climate change is a serious global health threat that has an impact on young people’s lives and may influence their mental health. Since the global climate strike movement, many adolescents have expressed worries about climate change. Thus, the aim of this study is to examine the prevalence of worries about climate change, and factors associated with worries about climate change, in a representative sample of Norwegian adolescents. Data were retrieved from Ungdata, an annual nationwide online youth survey. Adolescents (n = 128,484) from lower and upper secondary school participated in the study. Data were analysed descriptively and with logistic regression. Most of the adolescents were not worried or a little worried about climate change. Girls, pupils who had at least one parent with higher education and pupils from urban areas were more inclined to worry about the climate. Adolescents who worried about the climate had more symptoms of depression than those who were less worried. While worry about climate change may constitute an additional burden for adolescents experiencing depressive symptoms, such worry can also be seen to reflect climate-friendly values.

## 1. Introduction

Climate change is one of the most serious global health threats of the twenty-first century [1]. In recent years there has been a growing awareness of climate change among the general population [2,3], which has led to high media coverage [4]. The concept of climate can be defined as the finite temperature distribution over time that arises under the regime of varying external conditions [5]. Climate change involves not only increased frequency and severity of extreme weather events but also more ongoing and harmful changes, including sea-level rises, food insecurity and reduced availability of clean water [6]. Recent years have seen an increase in worries about air pollution and environmental damage among young people [7]. Air pollution has been associated with anxiety, more frequent use of mental health services and poorer general well-being [8]. A warmer climate can also lead to heightened anxiety and uncertainty, which in turn may negatively affect social relationships and attitudes towards other people [9]. Climate change has an impact on the lives of children and young people [9]. Studies have shown that both acute events due to climate change such as floods or droughts and longer-term changes (e.g., droughts causing erosion over time) may have a negative influence on mental health [10]. Evidence suggests that direct exposure to climate change events could lead to a series of mental health conditions such as depression [11], anxiety, stress, insomnia [12,13] and substance abuse [6]. Natural disasters caused by climate change affect individuals’ mental health [14], and acute severe climate events, like the heatwave in the summer of 2022 in Europe [15], might be traumatic. Anxiety and depression can be a consequence of these events, which again can lead to suicidal and risky behaviour, feelings of abandonment, and may even impact physical health [11]. Some studies found that a concern about climate change, combined with worry about the future, could lead to fear, anger, feelings of powerlessness, exhaustion, stress and sadness, which are often referred to as climate anxiety [14] and may be associated with increased substance abuse [16] and suicidal behaviour [17]. Research also indicates that anxiety about climate change is more prevalent among adolescents and young adults [2,18].

The growing awareness of the negative impact of climate change on mental health, especially among children and adolescents, has not only been in focus in research, but also in the media [19]. Research and the media often use the term climate anxiety [2,12,20]. The American Psychological Association defines anxiety as “an emotion characterized by feelings of tension, worried thoughts and physical changes like increased blood pressure” [9]. Anxiety is related to fear that alerts us to danger [2], whereas worry can be seen as a complex emotion derived from fear. Worry is an emotional experience that arises alongside repetitive unpleasant thoughts about the future and involves a focus on future negative events for which one can prepare [21]. Worry might lead to adaptive behaviour when the situation is seen as controllable. In contrast, if the situation is uncontrollable and there are extreme levels of worry, it may lead to stress and low well-being, and these negative consequences may at times outweigh the possible benefits of worries [12,21]. Overall, anxiety and worry may be related constructs [22] and climate change is a threat which affects the future of human beings. If worry about climate change is a key factor in one’s life, it can affect mental health [23]. Being worried about something may make a person realize the importance of taking action to prevent an unwanted outcome [21]. 

In a Norwegian study, people under the age of 30 years reported willingness to do something to improve the climate, such as reducing their flying and car use and adopting a more sustainable diet [4]. Another study showed that a large proportion of Norwegian adults were worried about climate change; here, women were more worried than men and those with higher education were more worried than those with lower education. Adults living in urban areas expressed more worry about climate change than those living in rural areas [19]. Despite these worries about climate change, Norwegian adults have been found to be less worried about climate change than adults in other European countries such as the UK, Spain or Italy [24]. 

Some studies have found a positive association between climate change, mental health, and substance abuse among adults [6]. However, do these patterns of worries about climate change also apply to adolescents? Based on prior research, we hypothesize a high prevalence of worries about climate change among Norwegian adolescents. Further, we assume that concern about climate change might increase cannabis and alcohol use and lead to more symptoms of depression. Our study addressed a large population of Norwegian adolescents and focused on worries about climate change, in contrast to previous research exploring climate anxiety. Thus, the aim of the study was to examine the prevalence of worries about climate change, and factors associated with worries about climate change, in a representative sample of Norwegian adolescents. 

## 2. Materials and Methods

### 2.1. Design and Study Population

The data used in this study came from Ungdata, the Norwegian Youth Survey conducted by the Norwegian Social Research Institute (NOVA) [25]. Ungdata is a self-administered, ongoing representative survey, targeting adolescents throughout Norway. The online questionnaire comprises a compulsory section with 159 questions for lower secondary school pupils (13–15 years) and 168 questions for those in upper secondary school (16–19 years) and covers various topics in adolescents’ lives, such as physical and mental health, life satisfaction, thoughts about the future, alcohol and drug consumption, leisure, and relationships. Since 2020, the questionnaire has assessed worries about climate change. Ungdata is offered to all municipalities and county councils throughout the country, and the questionnaire is administered in collaboration with NOVA and the Regional Drug and Alcohol Competence Centres (KoRus). Usually, the survey is conducted every third year in each municipality, but which municipalities perform the survey varies from year to year. Over a three-year period, most municipalities have conducted the survey at least once. All data are stored in a national database [25,26]. 

In 2021, the survey was completed in the majority of secondary schools in Norway. Participants in the survey were 139,841 pupils from lower secondary (13–15 years of age) and upper secondary schools (16–19 years of age) in 209 out of 356 Norwegian municipalities. The response rate was high: in lower secondary schools 83% and in upper secondary schools 67% [26]. Our sample is demographically comparable to Ungdata samples from previous years and is thus seen as representative of Norwegian adolescents. We excluded 11,357 (8.1%) pupils because of missing data on worries about climate change. The sociodemographic composition of the excluded group was similar to that of the included sample. Thus, data on 128,484 pupils were included in the final analysis.

### 2.2. Measures

#### 2.2.1. Outcome Variable

The pupils were asked to state to what extent they were worried about climate change with the response options ‘not worried at all’, ‘a little worried’, ‘quite worried’, ‘very worried’ and ‘I don’t know’. There were 2.4% of boys and 2.6% of girls who indicated that they did not know (N total = 5768). For the purpose of logistic regression, the variable was dichotomized into ‘not/a little worried’ versus ‘quite/very worried’, and pupils who ticked that they did not know were considered as missing values. This practice is in line with similar studies measuring worries about climate change [2,27].

#### 2.2.2. Independent Variables

##### Sociodemographics

We included the following five sociodemographic variables in the analysis: (1) gender (male/female; for statistical reasons, pupils who ticked “other” were coded with a missing value on the gender variable), (2) grade, which we used as an indicator for age, as age is not assessed by the survey, and (3) self-perceived family finances, which were assessed as a measure of socioeconomic background [28,29]. Parental education (4) was measured by asking the pupils whether 1 = none, 2 = one or 3 = both parents had higher education. For the logistic regression, this variable was dichotomized into ‘no higher education’ and ‘higher education’ (at least one parent with higher education). The measure of “centrality” (5), as defined by Statistics Norway [30], was used as a proxy for how centrally located a municipality in Norway is. The measure is based on how many jobs and service institutions can be reached by car within 90 min from where one lives. All municipalities are categorized on a scale from 1 to 6, were 1 indicates the most central municipalities and 6 the least central municipalities.

##### Mental Health

In Ungdata, the Depressive Mood Inventory was used to assess symptoms of depression. This measure was derived from the Hopkins Symptom Checklist and comprised six questions where pupils were asked to indicate whether they had been affected by any of the following during the past week: (1) Felt that everything is a struggle, (2) Had sleep problems, (3) Felt unhappy, sad or depressed, (4) Felt hopeless about the future, (5) Felt stiff or tense, and (6) Worried too much about things. All items had four response options: 1 = not been affected, 2 = been affected a little, 3 = been affected quite a lot, and 4 = been affected a lot. The mean scores were computed and dichotomized using a cut-off value of 3 or above for a pupil with a high degree of symptoms of depression, while those whose average was below 3 had a low degree of symptoms of depression. This procedure has been adopted in previous studies using Ungdata and it has been reported that the six items measuring depressive symptoms have acceptable reliability (Cronbach’s α =  0.88) [31]. Single item means were used for the description of the sample whereas the scale score for symptoms of depression was applied in multivariate analysis [32]. Further, pupils were asked to indicate whether they thought they would have a good and happy life in the future, with the response options 1 = yes, 2 = no and 3 = I don’t know. In the logistic regression, the latter two categories have been merged. Ungdata applies a modification of the “Cantril ladder” as adopted in the Gallup World Poll [33] as a measure of subjective well-being. Participants were asked to rate their level of well-being at the present time on a scale from 0 to 10 where 10 means the best possible well-being. A cut-off score was chosen at 6 and above for good perceived subjective well-being [34].

##### Leisure Activities

Ungdata measures screen time with the item ‘Outside school, how much time do you normally spend on activities that involve looking at a screen (TV, computer, tablet, mobile) each day?’ with the seven response options ranging from no time to more than six hours. Social media use was measured by asking ‘Think about what you do on a normal day. How much time do you spend on social media?’ (Response options: 1 = no time to 6 = more than 3 h). Based on previous studies, these items were dichotomized into ≤3 h versus >3 h for screen time [35] and ≤1 h versus >1 h for social media use [36]. Organized leisure time was assessed with the item ‘Are you currently a member of any organization, club, society or association or have you previously been a member of one since you were 10 years old?’ (Response options: 1 = yes, 2 = no, but previously, 3 = no, never).

##### Cannabis Use and Alcohol Intoxication

Cannabis use and alcohol intoxication were assessed by asking the participants if they had used cannabis or had been intoxicated by drinking alcohol in the previous 12 months. Possible responses were ‘never’, ‘once’, ‘2–5 times’, ‘6–10 times’, and ‘11 times or more’ and were dichotomized as in previous studies into ‘one or more intoxication episodes’ versus ‘no intoxication episodes’ for alcohol intoxication [37] and ‘no’ versus ‘one or more times’ for cannabis use [38].

### 2.3. Statistical Analysis

To determine the prevalence of worries about climate change, summary statistics were calculated for all variables overall and grouped by worries about climate change using the Chi-square test and Fisher’s exact test where appropriate. Multivariate logistic regression analysis was then applied to assess associations between the sociodemographic variables, mental health and leisure activities and worries about climate change. Grade (age) and centrality were treated as continuous variables, perceived family finances were coded as a dummy variable to fit into the logistic regression models, and all other independent variables were computed as binary as described above. The first (low) category of the independent variables was set as reference to simplify interpretation. Crude odds ratio (model 1) and adjusted odds ratios (models 2 and 3) were reported as measures of the association between worry about climate change and the independent variables [39]. In model 2 we adjusted for sociodemographic variables and leisure activities, whereas in model 3 we also controlled for mental health aspects, alcohol intoxication and cannabis use as potential confounders. The Omnibus test for model coefficients was significant in each of the three models, thus logistic regression analysis was considered as appropriate. The total explained variance of the dependent variable in each model was indicated by both Cox and Snell R2 and Nagelkerke R2. The test for the multicollinearity of the predictor variables was negative (VIF < 3). Statistical significance was set at *p* < 0.05. Data analysis was conducted using IBM SPSS Statistics for Windows (IBM Corp, Version 26.0, released 2016, Armonk, NY, USA).

### 2.4. Ethics

In this study, we analysed data from an already established data set, the Ungdata survey. Data were collected by the Norwegian Social Research institute (NOVA) in collaboration with the Regional Drug and Alcohol Competence Centres (KoRus). The Ungdata survey was administered anonymously online during school hours with a teacher or a public health nurse present. The pupils were informed that data collection was based on informed consent and that participation was voluntary. All parents were informed prior to the study (a passive consent scheme) and could refuse to let their children participate. This in line with Norwegian regulations on privacy protection. Permission to access and use the data was given by NOVA at Oslo Metropolitan University. The study was approved by the data protection office of Inland Hospital Trust with reference number 18778329. All methods were performed in accordance with applicable laws, regulations and research ethics guidelines.

## 3. Results

### 3.1. Prevalence of Worries about Climate Change

The prevalence of climate change worry among adolescents aged 13 to 19 years was 37.6%. All characteristics of the study population by worries about climate change are presented in Table 1. Worries about climate change were most prevalent in the 10th grade in lower secondary (15–16 years of age). Young people living in central areas were more worried about climate change than those living in less central areas. Young people who perceived that their family’s finances were poor were slightly overrepresented among those who worried about climate change. Within the group who were quite/very worried about climate change, most had at least one parent with higher education. Among those who were quite/very worried about climate change, slightly more were involved in organized leisure activities, whereas fewer spent more than one hour daily on social media. Differences in daily screen time between those who were worried and those who were not worried were not statistically significant.

Symptoms of depression above the cut-off were more prevalent among the young people who were quite/very worried about climate change. Among those who were quite/very worried about climate change, a lower percentage had high perceived well-being. Pupils who were quite/very worried about climate change were also more inclined to expect not to live a happy life than those who were not, or a little, worried. Further, the results showed that cannabis use was somewhat higher among those who were quite/very worried about climate change than among those who were not/a little worried, whereas the prevalence of any alcohol intoxication episodes was slightly lower among those who were quite/very worried.

As shown in Figure 1, among those who were quite or very worried, there were more females than males. Figure 2 indicates that when stratified by gender, girls in urban areas were nearly 2.5 times more often quite/very worried about climate change than boys living in rural areas.

### 3.2. Associations between Sociodemographic Factors, Leisure Activities, Mental Health, Cannabis Use and Alcohol Intoxication, and Worries about Climate Change

All logistic regression models are presented in Table 2. The unadjusted logistic regression (model 1) shows a significant association between worries about climate change and all independent variables, except daily screen time. Female gender, higher grade, neither poor nor good perceived family finances, having two parents with higher education, being engaged in an organized leisure activity, high level of depressive symptoms, and cannabis use were associated with higher odds of worries about climate change. Lower centrality (i.e., higher centrality scores), good perceived family finances, spending more than one hour daily on social media, higher subjective well-being, expecting to live a happy life, and having been intoxicated with alcohol were associated with lower odds of worries about climate change. There was a positive association between worries about climate change and depressive symptoms.

When including sociodemographic factors and leisure activities in model 2, daily screen time still remained as a non-significant factor. However, taking part in organized leisure activities and spending less than one hour daily on social media were associated with being quite/very worried about climate change. After adding mental health factors, cannabis use and alcohol intoxication as shown in model 3, the odds related to organized leisure activity increased. Compared to the crude odds ratios, the odds ratios related to depressive symptoms and alcohol intoxication decreased, while those related to subjective well-being, future life expectations and cannabis use increased. The odds of being worried about climate change increased by 10% if pupils had used cannabis, assuming the remaining variables are held constant. There were also increases in these odds for being engaged in organized leisure activities (21%), and for having depressive symptoms above cut-off (71%).

## 4. Discussion

The aim of this study was to examine the prevalence of worries about climate change, and factors associated with worries about climate change, in a representative sample of Norwegian adolescents. The results showed that almost 40% were worried about climate change, and that girls, pupils who had at least one parent with higher education and pupils from urban areas were more inclined to worry about the climate. The other important finding was that adolescents who were worried about climate change had more symptoms of depression than those who were not so worried.

The prevalence of worries about climate change, as found in our study, was lower than that reported in a recent international study on climate anxiety among children and young people in ten low-, middle- and high-income countries [2]. Hickmann et al. found that about 60% of respondents reported feeling “very” or “extremely” worried about climate change, and nearly half (45%) stated that their feelings about climate change were negatively affecting their daily life [2]. Discrepancies in prevalence may be due to other factors than the scale used when assessing climate worry, since the researchers merged the two highest categories, as we did in our study. In countries such as India, the Philippines and Brazil where the impact of climate change is more noticeable, the prevalence of worries was higher than in countries in temperate zones, such as Finland and the UK. However, the latter countries still had somewhat higher prevalence than our figures, with 44% extremely or very worried about climate change [2]. A lower prevalence of climate worry in our sample may therefore reflect a still modest impact of climate change in Norway, as the effects of climate change are hardly visible and do not affect individuals directly in their daily lives [40]. The low prevalence of worries about climate change among Norwegian adolescents is in line with the Norwegian adult population, which is among the least concerned about climate change in Europe [24,41]. This might be due to Norwegians having a high level of trust in government and the mentality that the state will take care of its citizens whatever happens [42], which was also observed during the COVID-19 pandemic [43,44].

The fact that girls are more worried than boys might be due to a higher general anxiety level among girls, which has been found in other studies [45]. Alternatively, or in addition, more worry about climate change among girls may be due to their greater factual knowledge about climate change [46]. Moreover, and despite the generally greater interest in political issues among men [47], women have been found to display greater support for environmental protection than men [48]. Provided such gender differences extend to the younger age groups, they may help to explain why girls were found to be more worried about the climate than boys.

An interesting finding is that the peak age of worry is the 10th grade of lower secondary school (15–16 yrs.). This concurs with other studies showing that youth perceptions and concern about climate change declined from younger to older youth and then rose again in young adults [49]. This “adolescent dip” in environmental attitudes might be due to the psychological and emotional development of the brain through adolescence with a progressive maturation from early adolescence to middle and late adolescence, which is the final phase of the organization of the adult brain close to young adulthood [50].

While perceived family finances were unrelated to climate worry, having at least one parent with higher education was related to a greater likelihood of worrying. Higher levels of education may logically translate into greater interest in and awareness of the state of the world, including climate change, and parents’ awareness is a major influence on children’s attitudes and behaviour related to environmental protection [51]. Thus, there is a logical link between higher levels of knowledge and awareness among parents, as indicated by their educational level, and their children’s emotional response to possessing that knowledge. While higher educational levels and wealth are often intrinsically related [52], our study indicates that children’s and adolescents’ worries about climate change may in part be viewed as cultural inheritance from their parents, while unrelated to their parents’ economic situation. In addition, the numbers of people with higher education in urban areas are higher than in rural areas, due to a greater density of service infrastructure which requires qualified and highly educated staff [53]. This may explain why our findings show a higher prevalence of worries about climate change in more central areas.

The study showed that having depressive symptoms above the cut-off was associated with greater likelihood of worrying about climate change. A question that remains is whether there are reasons not to worry about climate change. Is the lack of worry a form of denial or do those that worry actually possess personality traits such as high levels of neuroticism [45]. Another reason might be a high correlation between symptoms of depression and worries about climate change, so that worries about climate change become the main content of the rumination typical of depression. There might also be reciprocal forces in this relationship, whereby worries about climate change and depressive symptoms might be mutually reinforcing.

Anxiety related to the global climate crisis and environmental disaster has been found to be associated with panic attacks, insomnia, obsessive thinking and substance use disorders in addition to general anxiety and depression [54]. This is in line with our study, although we only measured symptoms of depression. We also found that there were differences in alcohol and cannabis use, as those who worried most were less likely to have been intoxicated with alcohol but more likely to have used cannabis.

Adolescents might be more likely than adults to experience additional effects associated with worries about climate change, such as negative future expectations or a lower perceived level of well-being. Adolescents are at a crucial point in their physical and psychological development, with enhanced vulnerability to the effects of stress and everyday anxiety [55]. There are, however, also indications that adult primary care patients tend to be concerned about climate change which also showed a positive correlation with dysphoria [56]. On the other hand, if we apply the description of ‘worry’ according to the research of Sweeny and Dooley, who defined worry as a motivator [21], we may interpret being worried about the climate as something positive. This could explain the high percentage of young people who are worried about climate change and are engaged in organized leisure activities. Thus, their worry does not paralyze, but activates them. This activism may lead as a next step to behaviour change to reduce climate change. As this assumption is vague, it requires further research in the form of qualitative studies.

### Strengths and Limitations

A strength of the present study is the large sample size with a high response rate. This high response rate reduces the possibility of selection bias. Data used in this study were collected in 2021 and provide an up-to-date description of Norwegian adolescents in lower and upper secondary school. Data were collected in 209 municipalities that cover a large area in Norway and have varying degrees of centrality.

The data used are self-reported, which might be a limitation. There is a risk that adolescents do not answer honestly or misunderstand questions. The questions had different response periods, ranging from one day to 12 months. This may give rise to errors because it is easier to remember one day than 12 months back. Several of the variables are dichotomized, which could lead to a lack of variation between individuals. Another limitation is that Ungdata is a cross-sectional study that precludes inferences about causal relationships.

## 5. Conclusions

The adolescents in this Norwegian study were worried about climate change, although most were not greatly worried. Worries about climate change were most common among girls, 15–16-year-olds and adolescents living in urban areas. The results indicate a real association between worries about climate change and mental health issues, but the causal relationship needs further studies. While worry about the climate may constitute an additional burden for young people with depressive symptoms, such worry can also be seen to reflect climate-friendly values. Under the right circumstances, such values may be channelled into climate-protective action. Drawing on the productive potential of climate worry, while minimizing the mental health burden associated with it, should be a priority in the coming years.

## Figures and Tables

**Figure 1 ijerph-19-12826-f001:**
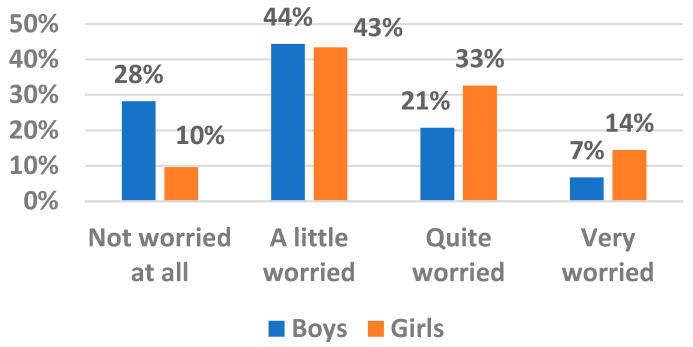
Prevalence of worries about climate change by gender.

**Figure 2 ijerph-19-12826-f002:**
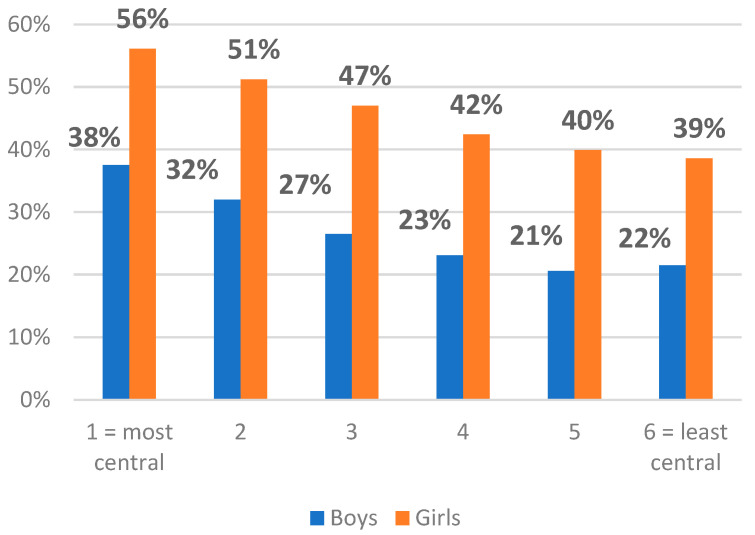
Prevalence of being ‘quite/very’ worried about climate change by gender and centrality of municipality of residence.

**Table 1 ijerph-19-12826-t001:** Characteristics of the young people by presence and absence of worries about climate change.

Variables	Not/a Little Worried N Total 75,436 (62.4%)	Quite/Very Worried N Total 45,273 (37.6%)	*p*-Value
	N (%)	N (%)	
Sociodemographic variables			
Gender Female Male	32,324 (43.6) 41,789 (56.4)	28,635 (64.6) 15,708 (35.4)	<0.001 *
Grade (age) 8th grade, lower secondary school (13–14 yrs.) 9th grade, lower secondary school (14–15 yrs.) 10th grade, lower secondary school (15–16 yrs.) 1st grade, upper secondary school (16–17 yrs.) 2nd grade, upper secondary school (17–18 yrs.) 3rd grade, upper secondary school (18–19 yrs.)	15,052 (20.6) 14,901 (20.4) 14,240 (19.5) 12,341 (16.9) 10,224 (14.0) 6140 (8.4)	8231 (18.6) 8434 (19.1) 8827 (20.0) 7361 (16.6) 6241 (14.1) 5144 (11.6)	<0.001 ^#^
Centrality 1 = most central 2 3 4 5 6 = least central	8419 (11.2) 14,895 (19.8) 29,467 (27.1) 16,979 (22.5) 11,049 (14.7) 3577 (4.7)	7619 (16.8) 10,819 (23.9) 12,143 (26.8) 8367 (18.5) 4816 (10.6) 1570 (3.5)	<0.001 ^#^
Perceived family finances Poor Neither poor nor good Good	2748 (3.7) 11,064 (14.9) 60,433 (81.4)	1877 (4.2) 7096 (15.9) 35,705 (79.9)	<0.001 *
Parental higher education None of the parents One parent Both parents	12,153 (17.2) 23,720 (33.6) 34,629 (49.1)	5110 (11.8) 12,584 (29.2) 25,432 (59.0)	<0.001 *
Leisure activities			
Organized leisure time No Yes	35,238 (47.3) 39,266 (52.7)	19,827 (44.2) 25,075 (55.8)	<0.001 *
Daily screen time ≤3 h >3 h	18,040 (24.2) 56,563 (75.8)	10,907(24.3) 34,021 (75.7)	0.712 ^#^
Daily social media time ≤1 h >1 h	19,153 (25.7) 55,498 (74.3)	12,178 (27.1) 32,737 (72.9)	<0.001 *
Mental health			
Depressive symptoms Low (<3) high (≥3)	61,501 (84.7) 11,131 (15.3)	33,028 (75.4) 10,795 (24.6)	<0.001 *
Subjective well-being <6 ≥6	12,792 (17.1) 62,193 (82.9)	9733 (21.6) 35,397 (78.4)	<0.001 *
Future life expectation I don’t expect to live a happy life I expect to live a happy life	21,814 (29.5) 52,212 (70.5)	15,855 (35.6) 28,706 (64.4)	<0.001 *
Cannabis use and alcohol intoxication			
Used cannabis previous 12 months No One or more times	69,447 (93.0) 5255 (7.0)	41,605 (92.5) 3364 (7.5)	0.004 ^#^
Alcohol intoxication previous 12 months No intoxication episode One or more intoxication episodes	51,262 (68.5) 23,576 (31.5)	31,253 (69.4) 13,803 (30.6)	0.002 ^#^

^#^ Fisher’s Exact Test. * Chi-square.

**Table 2 ijerph-19-12826-t002:** Logistic regression models with “worries about climate change” as the dependent variable.

	Model 1	Model 2	Model 3
	OR (95% CI)	aOR (95% CI)	aOR (95% CI)
Gender (ref. male)	2.35 (2.30–2.41) *	2.60 (2.53–2.67) *	2.47 (2.40–2.54) *
Grade (age)	1.06 (1.05–1.07) *	1.08 (1.07–1.09) *	1.12 (1.11–1.13) *
Centrality ¹	0.84 (0.83–0.85) *	0.84 (0.83–0.85) *	0.85 (0.84–0.86) *
Perceived family finances (ref. poor) Neither poor nor good Good	1.07 (1.04–1.11) * 0.90 (0.88–0.93) *	0.89 (0.82–0.96) * 1.26 (1.18–1.35) *	0.97 (0.90–1.05) 0.93 (0.87–1.00)
Parental education (ref. no higher education)	1.55 (1.49–1.60) *	1.55 (1.49–1.61) *	1.57 (1.52–1.61) *
Organized leisure time (ref. none)	1.13 (1.10–1.16) *	1.17 (1.14–1.20) *	1.21 (1.18–1.25) *
Daily screen time (ref. < 3 h)	0.99 (0.96–1.02)	1.01 (0.97–1.04)	0.97(0.93–1.00)
Daily social media time (ref. < 1 h)	0.92 (0.90–0.95) *	0.67 (0.65–0.69) *	0.66 (0.64–0.71) *
Depressive symptoms (ref. low (<3))	1.80 (1.75–1.86) *		1.71 (1.10–1.42) *
Subjective well-being (ref. < 6)	0.75 (0.73–0.77) *		0.91 (0.88–0.95) *
Future life expectation (ref. no happy life)	0.75 (0.73–0.77) *		0.87 (0.85–0.90) *
Cannabis use (ref. none)	1.07 (1.02–1.12) *		1.10 (1.04–1.16) *
Alcohol intoxication (ref. none)	0.96 (0.93–0.98) *		0.79 (0.74–0.81) *

Model 1: Crude Odds Ratio (OR); Model 2: Adjusted for sociodemographics and leisure activities; Cox and Snell R^2^: 0.07, Nagelkerke R^2^: 0.09; Model 3: Adjusted for sociodemographics, leisure activities, mental health, cannabis use and alcohol intoxication; Cox and Snell R^2^: 0.07, Nagelkerke R^2^: 0.10; ^1^ Higher is less central, * *p* < 0.05.

## Data Availability

Data is available from the Norwegian Centre for Research Data (NSD) on request.

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
