# Peer review of "Associations between Mental Health, Lifestyle Factors and Worries about Climate Change in Norwegian Adolescents"

_ijerph, 2022, doi:10.3390/ijerph191912826_

Round 1
Reviewer 1 Report
The manuscript “Associations between mental health, lifestyle factors and worries about climate change in Norwegian adolescents” is a cross-sectional study investigating associations between different individual background and lifestyle variables and the likelihood of being worried about the climate change among adolescents in Norge. The topic is relevant and in general the study is conducted in a correct manner. It is mainly the background and the presentation of the results that needs a better structure.
Abstract:
Line 12-15: It is not clear at all what the difference is between “climate anxiety” and “worries about climate change” here, so the aim becomes very confusing. I would suggest that you skip mentioning climate anxiety in the abstract and just focus on “worries about climate change”.
Introduction:
In their background discussion, the authors do not really differ between the topics of primary and secondary effects of climate change on mental health. Being worried about climate change as such is quite different from developing a PTSD because one has been affected by a climate catastrophe (and the last one is not a subject of this manuscript at all). Thus, the background would need to be a bit more focused on the topic.
Line 47-49: Yes, it is completely true that “Air pollution has been associated with anxiety, more frequent use of mental health services and poorer general well-being” however, it is not clear, why the effects of air pollution are suddenly mentioned here.
Line 57-70: This is a very important paragraph as it is supposed to clarify the difference between worry and anxiety. I personally, however, became more confused after reading this than I was before. I would suggest that the paragraph is restructured so this important difference becomes clearer. Furthermore, I miss the part where the authors declare why the assessed outcome of this study was “worry” and not “anxiety”.
Methods:
Line 104-105: Were there any clear differences between the groups of the 8.1% excluded and the included students regarding socioeconomic factors and the place of residence (urban, rural)?
Line 11-113: I understand the need to dichotomize the data for logistic regression. However, this “I do not know” group is rather peculiar, and it would be interesting to know how it differs from the “yes” and “no” groups.
Results:
Please refer to a figure or table every time you mention a specific result in the text – right now the beginning of the Results parts lists the results depicted in Table 1 and in Figs 1 and 2 in a kind of “higgledy-piggledy” manner, so sometimes it is not clear if the authors are still talking about data in Fig 1 or if they are back referring to Table 1.
Figure 1 and 2: I miss some kind of variation indicator on the staples.
Lines 237-239: As this is a cross-sectional study, the authors should talk about associations between the exposure and outcome, rather than about causality (i.e., “worries about climate change increase the probability for depressive symptoms” and “worries about climate change reduce the likelihood for low level of subjective well-being”).
Discussion:
I would suggest that the authors change “was not or only little worried about climate change” to “was not or was only little worried about climate change”.
Line 311-312: Was there a high correlation between higher education and the degree of urbanization in your sample?
Line 351: Using wording “…is not that high among Norwegian adolescents” kind of implicates a comparison so maybe you should continue this sentence with “…is not that high among Norwegian adolescents than among …”
Author Response
Dear reviewer 1, many thanks for your constructive suggestions on how to improve the manuscript. In the attachment you will find our point-by-point responses to your questions and concerns. All major changes are highlighted in the revised manuscript in blue. We have additionally consulted an English native speaker to check the manuscript.

Reviewer 2 Report
Please see the attached file.

Author Response
Dear reviewer 2,
Many thanks for your constructive suggestions on how to improve the manuscript. In the attachment you will find our point-by-point responses to your questions and concerns. All major changes are highlighted in the revised manuscript in blue. The manuscript has now been checked by an English native speaker.

Round 2
Reviewer 1 Report
I am satisfied with the responses from the authors.